# The Academic Collaborative Center Older Adults: A Description of Co-Creation between Science, Care Practice and Education with the Aim to Contribute to Person-Centered Care for Older Adults

**DOI:** 10.3390/ijerph17239014

**Published:** 2020-12-03

**Authors:** Katrien Luijkx, Leonieke van Boekel, Meriam Janssen, Marjolein Verbiest, Annerieke Stoop

**Affiliations:** Department Tranzo, Tilburg School of Social and Behavioral Sciences, Tilburg University, P.O. Box 90153, 5000 LE Tilburg, The Netherlands; L.vanBoekel@tilburguniversity.edu (L.v.B.); M.Janssen4@tilburguniversity.edu (M.J.); M.E.A.Verbiest@tilburguniversity.edu (M.V.); H.J.Stoop@tilburguniversity.edu (A.S.)

**Keywords:** person-centered care, older adults, co-creation, perspective of older adults, quality improvement

## Abstract

Long-term care for older adults is in transition. Organizations offering long-term care for older adults are expected to provide person-centered care (PCC) in a complex context, with older adults aging in place and participating in society for as long as possible, staff shortages and the slow adoption of technological solutions. To address these challenges, these organizations increasingly use scientific knowledge to evaluate and innovate long-term care. This paper describes how co-creation, in the sense of close, intensive, and equivalent collaboration between science, care practice, and education, is a key factor in the success of improving long-term care for older adults. Such co-creation is central in the Academic Collaborative Center (ACC) Older Adults of Tilburg University. In this ACC, Tilburg University has joined forces with ten organizations that provide care for older adults and CZ zorgkantoor to create both scientific knowledge and societal impact in order to improve the quality of person-centered care for older adults. In the Netherlands, a “zorgkantoor” arranges long-term (residential) care on behalf of the national government. A zorgkantoor makes agreements on cost and quality with care providers and helps people that are in need of care to decide what the best possible option in their situation is. The CZ zorgkantoor arranges the long-term (residential) care in the south and southwest of the Netherlands. This paper describes how we create scientific knowledge to contribute to the knowledge base of PCC for older adults by conducting social scientific research in which the perspectives of older adults are central. Subsequently, we show how we create societal impact by facilitating and stimulating the use of our scientific knowledge in daily care practice. In the closing section, our ambitions for the future are discussed.

## 1. Introduction

Due to demographic changes and economic constraints, healthcare in general as well as long-term care for older adults are in transition. People prefer to participate in society and to live at home for as long as possible. Governments are pursuing agendas that seek to enable older adults to do so [1,2]. Enabling autonomy and community support and health services that are tuned to the needs and possibilities of older adults are key domains for the World Health Organization (WHO) in their aim to create age-friendly communities [3]. Due to ageing of the population and a presumed unattractiveness of the profession, shortages in caregiving staff in long-term care for older adults are already a fact [4,5]. To date, the large-scale implementation of technological solutions that aim to support older adults to lead the life they desire is rather an ideal than reality [6,7,8,9]. Moreover, over the last decades, the care for older adults, especially within nursing homes, has been shifting from the biomedical care model towards a person-centered care (PCC) model [10,11]. PCC is suitable for age-friendly communities, because both stress the importance of older adults to be valued, actively involved and supported when needed [12,13,14]. Nursing homes providing PCC would be excellent age-friendly living environments, according to the new smart age-friendly ecosystem framework [12]. PCC refers to the recognition and appreciation of the individuality of both care receivers (i.e., older adults) and caregivers. PCC aims to enable older adults with complex care needs to continue to live their lives as they prefer despite an increasing dependency on others [10,11,15,16,17]. To provide tailored care and support for older adults, sensitivity regarding their individual needs, wishes and possibilities is essential, but challenging for care professionals. There is an implementation gap; although care organizations embrace PCC, they struggle to bring it into practice [18].

In order to address the aforementioned challenges, organizations providing long-term care for older adults increasingly use scientific knowledge to evaluate and innovate long-term care [19,20,21,22]. The use of scientific knowledge to improve the quality of PCC for older adults incorporates the stimulation and facilitation of care professionals to reflect on and creatively and critically think about the care they provide. The education of current and future care professionals is one of the ways this can be achieved. Close, intensive and equivalent collaboration between science, care practice and education in the development of innovative, evidence-based knowledge—also called co-creation—is key in the success of improving long-term PCC for older adults by using scientific knowledge [19,20,21,22].

Such co-creation is central in the Academic Collaborative Center (ACC) Older Adults of Tilburg University in the Netherlands. This ACC is a long-term structural partnership of ten organizations providing long-term care for older adults, CZ zorgkantoor and Tilburg University (see Box 1 for the organizational embedding of the ACC Older Adults and Table 1 for an overview of the stakeholders involved). In the Netherlands, a “zorgkantoor” arranges long-term (residential) care on behalf of the national government. A zorgkantoor makes agreements on cost and quality with care providers and helps people that are in need of care to decide what the best possible option in their situation is. The CZ zorgkantoor arranges the long-term (residential) care in the south and southwest of the Netherlands. The ACC Older Adults chose a social scientific perspective for two reasons. First, the shift from the biomedical care model towards a PCC model underlines the fact that compared to medical care, living is at least equally important to nursing home residents. Second, Tilburg University has no medical school and therefore no medical expertise. Our slogan “science in practice to contribute to PCC for older adults” summarizes our ambitions:Creating scientific knowledge by contributing to the knowledge base on PCC for older adults by conducting social scientific research in which the perspectives of older adults are central;Creating societal impact by facilitating and stimulating the use of our scientific knowledge in daily care practice to enhance PCC.

Box 1Organizational embedding of the Academic Collaborative Center (ACC) Older Adults.
*ACC Older Adults*
The ACC Older Adults is a long-term structural partnership of ten organizations that care for older adults (Azora, BrabantZorg, De Riethorst Stromenland, De Wever, Groenhuysen, Schakelring, Surplus, SVRZ, Volckaert and Zorggroep West- en Midden-Brabant (Thebe)), CZ Zorgkantoor and Tilburg University. All partners invest both in-kind and financially in the ACC. Since 2018, the Dutch Ministry of Health, Welfare and Sport has structurally funded the ACC Older Adults to further develop and strengthen the knowledge infrastructure, i.e., improve and expand the collaboration and knowledge transfer between science and care practice in our ACC.
*Tranzo, Tilburg University*
The ACC Older Adults is embedded within Tranzo, which is one of the departments of Tilburg University in the Netherlands. Tranzo aims to bridge the gap between science and practice in different domains of care and welfare. In addition to the ACC Older Adults, Tranzo consists of ten other ACCs with different areas of interest, such as mental health care and people with an intellectual disability.
*National Embedding*
In addition to the ACC Older Adults in Tilburg, five comparable ACCs in the field of Care for Older Adults exist across the Netherlands. The other ACCs are affiliated with Maastricht University, Radboud University Medical Center Nijmegen, Leiden University Medical Center, Amsterdam University Medical Center—location VUmc and University Medical Center Groningen. Each ACC has its own specific expertise, ambition, approach and research focus in the field of care for older adults. On a national level, however, joined forces across the six ACCs add value to obtaining the shared goal of improving overall quality of care for older adults in agenda setting of the national government. For example, during the current COVID-19 outbreak, the six ACCs are cooperating in various corona-related studies in order to inform policymakers on a national level [23,24,25]. All six ACCs receive structural funds from the Dutch Ministry of Health, Welfare and Sport to strengthen the knowledge infrastructure.

This paper aims to describe how we are joining forces in the ACC Older Adults to realize our ambitions. Although the first ambition is mainly scientific and the second mainly societal, they are strongly interwoven. Co-creation is the key approach for realizing both ambitions and includes intensive and equal collaboration between various stakeholders operating on different levels, ranging from older adults, caregivers, nurses, psychologists, managers, teachers, policymakers, and scientists. The next section starts with a description of how we create scientific knowledge, because this is the starting point for all our activities. We elaborate on how we are co-creating with various stakeholders in order to ensure that our scientific results are relevant and usable for daily care practice. The following section subsequently elaborates on how we are creating societal impact based on our scientific results and insights.

## 2. Creating Scientific Knowledge

PCC is taking shape in the care relationship between older adults and care professionals. Therefore, when aiming to contribute to the knowledge base on PCC for older adults, it is important to address research topics that are of importance to older adults and care professionals and especially to discover the perspectives on these topics of older adults themselves. To ensure that we develop relevant and applicable knowledge, we are creating knowledge via co-creation in which older adults, care professionals, and researchers are involved. In the following sections, we elaborate on the specific themes of our studies, and on the stakeholders involved.

### 2.1. Themes of Research

Research projects in the ACC Older Adults—of which the majority are PhD studies—all relate to PCC and revolve around the following four themes: 1. autonomy; 2. technology; 3. social needs and social networks; 4. quality improvement.

Researchers, care professionals, and older adults together determine research topics, questions and designs for specific PhD studies, thus ensuring our research themes are of relevance for daily care practice. Partnership organizations facilitate data collection within their organization. Our PhD studies are financed in two ways, the first being competitive external research funding. Proposals for these PhD studies are always written in collaboration between researchers, care professionals of our partnership organizations, and older adults or their representatives. Research topics are determined based on gaps in scientific knowledge in combination with experiences of older adults and care professionals in care practice. These research projects are executed by full-time PhD students motivated to advance care practice with their scientific knowledge.

The second way PhD studies are financed is via funding by partnership organizations of one of their employees as a science practitioner. Science practitioners combine their PhD study with their current work as a care practitioner, for example, as a nurse, a psychologist, a manager, or a policy maker in a care organization. Their research topics relate to problems or questions they come across in their daily work, such as access to care [26,27,28], love, intimacy, and sexuality of nursing home residents with dementia [29,30,31,32], or autonomy [33,34]. Within their care organization, science practitioners are seen as experts in their field of research, and they also motivate their colleagues to use scientific knowledge. Moreover, care organizations explicitly connect to science by providing their science practitioner with the opportunity to conduct their PhD study, and they stress the importance of knowledge about the studied topic.

### 2.2. Stakeholders Involved

PhD students, both science practitioners and full-time PhD students, are supervised by the professor and one of the four senior researchers of the ACC. These senior researchers are employed by Tilburg University and hold a PhD degree. We aim to involve both scientific and societal expertise in our supervising teams. Therefore, if possible, an employee with a PhD degree of the care organization where the science practitioner is employed is involved as co-supervisor. This strengthens the collaboration between Tilburg University and the care organization. In addition to supervising PhD students, the four senior researchers of the ACC together with the professor further develop and expand the ACC Older Adults, prepare competitive research grant proposals, and also work partly as so-called “research brokers” within the partnership organizations to foster the collaboration and knowledge exchange between science and care practice. Each of them is responsible for the collaboration with two or three of the partnership organizations. In these organizations, the research brokers ensure that care practice and university are connected in various ways and at different levels. They meet and collaborate with people who are relevant for the use of scientific knowledge in daily care practice, they participate in relevant committees, and they know when and how to showcase the knowledge that is available in the ACC. The close connection that research brokers have with daily practice contributes to a detailed understanding of the struggles and interests of care professionals and older adults. This is essential for conducting science that is relevant and useful for daily practice.

Moreover, for each PhD study, we install an expert group of care professionals, managers and older adults. We regularly share and discuss our (preliminary) study findings and future study directions with this group to stimulate and facilitate care professionals in using our study findings in their daily work. Such an expert group ensures that insights from the study immediately are translated into daily care practice on a small scale. Additionally, due to the feedback of care professionals and older adults on the ongoing study, the study may be adapted to remain relevant and applicable for daily care practice.

### 2.3. Perspective of Older Adults

In order to be able to improve PCC by conducting scientific research and achieving its translation to care practice, it is essential to understand the perspectives of older adults. In our research, we always study the perspective of older adults on a specific topic, such as access to long-term care [26,35], technology acceptance [36,37,38,39,40], love, intimacy and sexuality [31,32], and social needs and networks [41,42,43,44]. Although physical or cognitive limitations—including dementia—may complicate this, we showed that it is possible and worthwhile [45]. The scarce number of studies that compare the perspective of older adults with that of proxies, such as loved ones or care professionals, shows that these perspectives differ [46,47,48,49]. These (sometimes nuanced) differences in perspective may affect the older adult’s experience of being respected as a unique individual and the genuine experience of PCC. To maximize the impact of understanding the perspective of older adults, we study it in the first empirical study of most research projects. Because older adults are a heterogeneous group regarding their capacities, limitations, goals, preferences, and habits, qualitative methods of data gathering are the most helpful for understanding the perspectives of older adults.

To clarify the value of studying the perspective of older adults, we give two examples. Repeated interviews with community dwelling older adults revealed that emotional attachment, need compatibility, cues to use, proficiency to use, input of resources, and support together influence the frequency of technology use. The interplay between these factors can be modified by factors, such as social influence, alternative means, and changing personal needs, which can cause changes in the frequency of technology use [39]. Interviews with older adults living with dementia in a nursing home and their spouses revealed that love, intimacy and sexuality are still important in their relationship despite the dementia and the move to a nursing home. However, the nursing home is not always conducive to satisfactory experiences regarding love, intimacy, and sexuality due to practical, emotional, and communicational problems and limitations [31].

To contribute to PCC, we aim to create an ACC with, for, and by older adults. Therefore, older adults need to be involved in the ACC on other levels as well. The research brokers aim to meet annually with the central client council of the partnership organizations to learn what is important to older adults and which topics we need to study, as well as to ask for feedback on our ambitions, research and products. Moreover, we are currently developing an expert panel of older adults to incorporate their perspectives structurally in the ACC. For instance, we invite older adults to share their opinions and perspectives on a certain topic during our symposia or events. These older adults like to share their story and to talk in public. In addition, we have a group of client representatives or members of special-interest organizations for older adults who are consulted in research proposals for funding opportunities and in ongoing research projects. These client representatives often have experience with a specific area such as policy, management or informal caregiving, depending on the topic of the project or research we invite them to collaborate on. Finally, we strive to set up a test panel of older adults from the general public who can pilot test our products and materials and provide us with feedback on this.

## 3. Creating Societal Impact

We aim to create societal impact by facilitating and stimulating the use of scientific knowledge in daily care practice to enhance PCC. Therefore, our study findings need to be translated in such a way that care professionals whom they concern can easily find, understand, and work with them. Moreover, with our scientific knowledge, we want to stimulate them to critically reflect on their care giving approaches and to inspire them to change these according to the study findings. Although dissemination is an important first step, it is also important to translate study insights into education and into practical tools that are ready to be implemented and used in daily care practice.

Our “science-to-practice team”, consisting of a communication expert, an education expert, and an implementation expert, plays a crucial role in creating societal impact and is complete as from the beginning of 2020. These team members work from their respective expertise in communication, education and care practice, and although they are not scientific staff, they understand and embrace the value of science to further improve care. Their mutual collaboration is close, and they collaborate closely with the researchers and other stakeholders in the ACC as well. These experts often notice other or earlier opportunities than our scientists to communicate or share research findings. Hence, working in such a multidisciplinary team with scientists as well as communication, education, and implementation experts is essential in creating both scientific knowledge and societal impact in order to strengthen PCC for older adults.

Moreover, care professionals are important stakeholders in creating societal impact because PCC needs to be realized in the care relationship between older adults and care professionals. Therefore, we incorporate the perspective of care professionals structurally in the ACC by having an expert team of care professionals. This expert team is consulted, for example, regarding the topics that need to be studied, the tools we develop based on the findings of our PhD studies, and the resources they use to find information. Furthermore, they may act as ambassadors for the ACC and help us to further disseminate and implement our knowledge among relevant stakeholders, such as their co-workers.

The following section shows how we translate our scientific knowledge into daily care practice in so-called “science-to-practice projects”. Subsequently, we elaborate on communication, education and implementation.

### 3.1. Science-to-Practice Projects

In so-called “science-to-practice projects”, in co-creation between researchers, older adults, care professionals, and teachers, we translate relevant scientific knowledge into daily care practice and education. Within these projects, based on our research findings, we develop tools and educational programs that can directly be applied in daily care practice, in vocational education, or for training on the job purposes. In general, these tools aim to help care professionals to analyze and evaluate their daily care giving practice and to pay attention to perspectives of older adults, which contributes to more PCC. An example of a “science-to-practice project” is the practical evaluation tool which is useful for care practice to evaluate and improve geriatric rehabilitation care [50]. This evaluation tool has been developed based on our previous scientific insights about evaluation of integrated care interventions and is described in detail elsewhere [51]. Another example is the Gerontechnologies Matchmaking (GTM) tool (see Figure 1) [52] that is based on our scientific insights about why older adults use or do not use digital technology [38]. The GTM tool was co-created with technology consultants and aims to help professionals to match the most appropriate technology at the right time with the needs of the older adult. Moreover, it structurally involves the perspective of informal caregivers, because they are important in the lives of older adults in general and regarding technology acceptance. The common boxes in Figure 1 depict objective information that is the same for the older adult and the informal caregiver, while the doubled boxes represent the topics in which the attitude and experience of both parties might differ. In the development of the GTM tool, we used our scientific insights about origins and consequences of technology acquirement by older adults and we observed the matchmaking dialogues between technology consultants and older adults. Together, these scientific and practical insights and observations were combined in the first version of the tool and thereafter drafted, tested and adapted. The approach has been described in detail elsewhere [52].

In some cases, the science-to-practice projects reveal undiscovered research areas and are the start of a new PhD study. For example, the PhD study on love, intimacy, and sexuality [29,30,31,32,45] revealed the perspective of residents and their loved ones on how nursing home residents with dementia, and possibly their partners, can be best supported in their wishes and needs with regard to intimacy and sexuality [53]. To translate the knowledge of this PhD study into daily care practice, the science-to-practice project started with semi-structured interviews with various nursing home professionals to discover their perspective. These interviews revealed that these care professionals usually perceive expressions of sexuality as problematic behavior and find it difficult to address the wishes and needs of residents with dementia with regard to love, intimacy and sexuality. To address the need of a training program to help care professionals to cope with all kinds of expressions of intimacy and sexuality, and to provide PCC with regard to such a sensitive topic, we started a new PhD study. In this PhD study, based on an understanding of the perspectives of both nursing home residents and care professionals, we aim to develop, implement and evaluate a comprehensive program to help care professionals cope with problematic behavior and to address needs and wishes regarding love, intimacy, and sexuality as well.

### 3.2. Communication

Communication is essential when aiming for societal impact, and it is interwoven in all activities to create societal impact. Our communication expert, who is employed by Tilburg University, with a background in marketing and communication, developed a marketing and communication strategy, including a recognizable corporate identity that is attractive to care professionals. Moreover, she formulated a slogan that summarizes our ambitions concisely and helps us focus: “science in practice to contribute to PCC for older adults”. We study our communication target group, professionals working in care for older adults, to understand how we can best reach them. In an attempt to overcome the implementation gap, we develop attractive and easy to understand personalized communication materials to inform care professionals about our study results and tools and to help them to change daily care practice according to our scientific knowledge.

The research brokers, the education expert, and the implementation expert use the communication materials as an aid to translate our scientific knowledge towards daily care practice. Moreover, our communication expert facilitates all our researchers in communicating in an attractive and accessible manner for an audience of care professionals. Furthermore, our meetings are interactive with the use of creative work forms, based on to the assumption that knowledge is more easily incorporated when offered through various approaches [54].

### 3.3. Education

The majority of professional caregivers in daily care for older adults do not have many years of education and is low or moderately skilled. Because educating current and future care professionals is an excellent means of translating scientific knowledge into daily care practice, middle and high vocational teachers as well as education experts of our partnership organizations have been involved in our ACC in recent years. Based on our research findings, our education expert, who is employed by Tilburg University and is an experienced teacher at both vocational levels, designs and tests education in co-creation with these stakeholders. For example, she designed, in close collaboration with a teacher from middle vocational education, a three-piece workshop based on the findings of our PhD study on love, intimacy and sexuality among nursing home residents [29,30,31,32,45]. The workshops have already been used in a number of classes, and we are now searching for opportunities to disseminate the tested and improved workshop “intimacy and sexuality” among teachers of vocational education on a national level.

We aim for education to be used by teachers and education experts themselves and aim for our scientific knowledge to be incorporated in the current curricula because we are convinced that this is most sustainable. Furthermore, we aim to spread our knowledge in care organizations and educational institutions across the Netherlands, without providing education by people employed by university. This ambition does not fit with the ambitions of a university department, in which research and educating academic students are rewarded, while teaching future and current caregivers and nurses is not. Moreover, other parties are experts in this, with whom we collaborate to realize our societal ambitions. We are convinced that this will lead to more sustainable results with which we can reach many more stakeholders.

Furthermore, in line with our ambitions, we organized a national inspiration day (see Figure 2) exclusively for (student) nurses and caregivers about love, intimacy and sexuality among nursing home residents. Together with teachers of higher vocational education, we designed and facilitated this day. We offered 100 (student) nurses and caregivers the opportunity to be actively involved and to learn about the topic. We invited them to design innovations for facilitating intimacy and sexuality in the nursing home in a person-centered way. All innovations were presented in an exposition. Visitors of the exposition selected the two most promising innovations: a board game (see Figure 3) and a leaflet. Both innovations are currently further fine-tuned, prototyped, and tested in care practice. This is conducted in co-creation between the ACC, the involved institution for higher vocational education, and the involved current and student professional caregivers. The board game aims to stimulate and facilitate conversation among team members regarding intimacy and sexuality in the nursing home. The leaflet aims to inform older adults who are going to live in a nursing home that not living together does not have to mean not being together—it is an invitation to bring this subject to the table. These products will become available for our partnership organizations, but also for other nursing homes in the Netherlands.

### 3.4. Implementation

Despite the aforementioned efforts and practical tools resulting from our science-to-practice projects, the utilization of these tools is not always obvious and easy from the care organization perspective. Therefore, we need to stimulate and facilitate implementation of our tools in daily care practice. To this end, our implementation expert, who is employed by Tilburg University as from the beginning of 2020, works on the cutting edge of care practice and science. With a background in management and quality improvement within care for older adults, she is familiar with the setting and challenges of adopting innovations. She offers our tools to our partnership organizations based on their demand within the organization. Moreover, she verifies whether an organization or a specific team is ready to use the tool, i.e., not hindered by daily chores, such as severe staff shortages, reorganization, or other organizational factors.

Although we aim for all Dutch nursing homes to use our tools, we do not introduce our tools in all nursing homes. In order to serve as much stakeholders as possible, our implementation expert trains self-employed professionals or nursing home professionals on how to implement our tools. To ensure the implementation of our tools is of high quality and performed as intended, she supervises and monitors the implementation process.

## 4. Future

The current state of the ACC Older Adults as described in this paper is characterized by a long-term, structural equal collaboration between various stakeholders, in which everyone’s expertise is respected and valued and mistakes are seen as opportunities to learn from. A clear focus and aim of the ACC and the research, which adheres to the mission and interests of stakeholders involved, is essential. Moreover, collaboration within multidisciplinary teams is important to address research topics that are relevant for care practice. It takes time over the years and every day to collaborate successfully on different levels and between different people and thus investments of all parties involved.

The ACC Older Adults was founded in 2003 as the ACC Chronic Care and made a focus shift towards PCC for older adults in 2013. It has always been our ambition to develop usable scientific knowledge, to translate this knowledge to daily care practice, and to facilitate its use. Structural funds of the Dutch Ministry of Health, Welfare and Sport to expand the knowledge infrastructure as from 2018 have provided us with the opportunity to enlarge the activities described above in recent years and will continue to do so in the future. We aim for our knowledge and tools to be translated and implemented into care practice. Co-creation is key in our approach. Combining the expertise of all stakeholders involved leads to the best possible outcomes for both science and care practice, and, thus, for PCC for older adults. For now, we focus on our partnership organizations, but we ultimately strive to reach all Dutch organizations providing long-term care for older adults and all institutions that educate professional caregivers in care for older adults. Therefore, we aim to use existing roads to disseminate our scientific knowledge, research findings, and tools, for example, by collaborating with institutes that share applicable knowledge or institutes that provide educational materials on a national level. We do not guide the implementation or education because we are convinced that it is more sustainable when we collaborate with other parties who are experts in implementation or education and are eventually able to serve all stakeholders across the Netherlands. Moreover, both implementing tools in care practice and vocational education do not fit the ambitions of a university department.

To date, our focus has mainly been on the development of scientific knowledge and evidence-based tools to implement in daily care practice and of education. In the future, when a tool has been implemented in different teams and organizations, we intend to evaluate the implementation processes and the impact on daily care practice and especially client outcomes. In this manner, we strive to strengthen our expertise in developing relevant scientific knowledge and in translating this knowledge into care practice to improve caregiving and especially to contribute to PCC and age-friendly communities.

## Figures and Tables

**Figure 1 ijerph-17-09014-f001:**
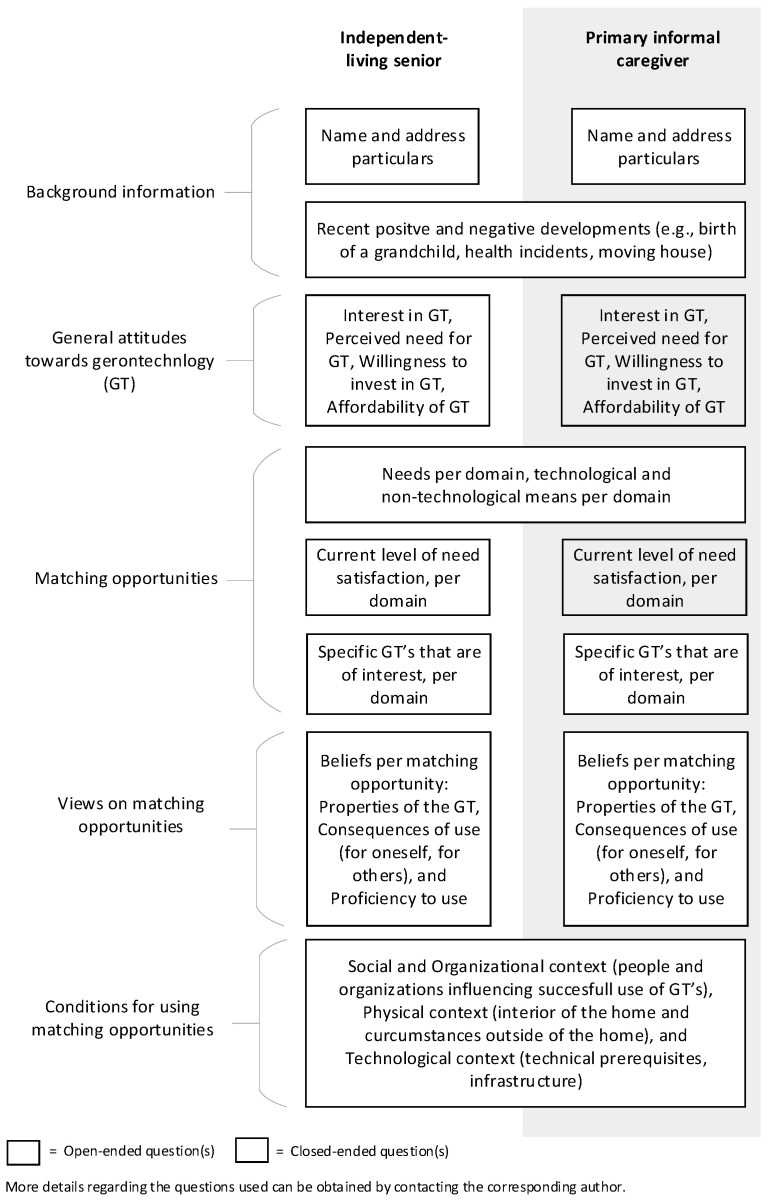
Gerontechnologies Matchmaking (GTM) tool [52].

**Figure 2 ijerph-17-09014-f002:**
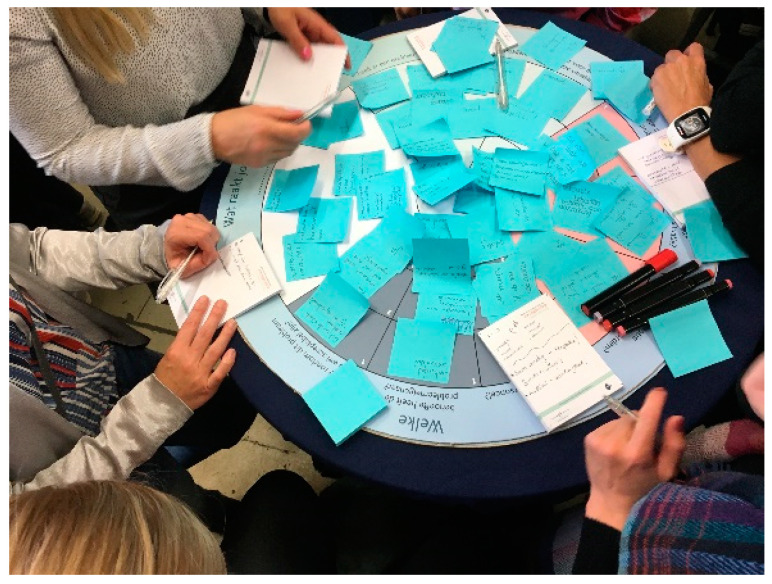
National inspiration day 2019.

**Figure 3 ijerph-17-09014-f003:**
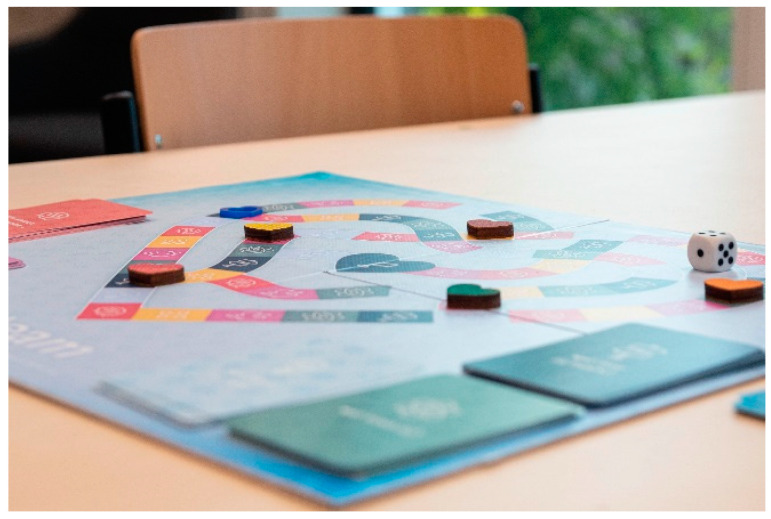
Board game to stimulate and facilitate conversation among team members regarding intimacy and sexuality in the nursing home.

**Table 1 ijerph-17-09014-t001:** Stakeholders involved in the ACC Older adults.

Stakeholders Involved	Employed by	Number in the ACC Older Adults (November 2020)	Background and/or Expertise	Main Tasks
Professor	Tilburg University	1	PhD degree in social sciences	Supervision of research projects and projects creating societal impact, chairing the ACC, strategy
Senior researcher/research broker	Tilburg University	4	PhD degree in social or health sciences, or psychology	Supervision of research projects and fostering collaboration and knowledge exchange between science and care practice and strategy
Communication expert	Tilburg University	1	Expertise in marketing and communication	Creating communication strategy and communication materials
Education expert	Tilburg University	1	Expertise in education, teaching	Fostering collaboration with education institution and making knowledge accessible in training and current curricula
Implementation expert	Tilburg University	1	Expertise in management and quality improvement	Supervising and monitoring implementation of our knowledge
Regular PhD student	Tilburg University	3	Various backgrounds	Conducting PhD research in collaboration with care practice and older adults
Science practitioner	Care organization and Tilburg University	7	Various backgrounds	Conducting research in collaboration with care practice and older adults
Science-to-practice project researcher	Tilburg University	1	Various backgrounds	Conducting research in collaboration with care practice and older adults
Post doc researcher	Tilburg University	3	Various backgrounds	Conducting research in collaboration with care practice and older adults
Care professionals	Care organization	Depending on the projects		Learning of and providing feedback to our studies, co-creating products, and providing feedback on our approach
Older adults	Not applicable	Depending on the projects		Provide their feedback about our knowledge development and spreading, being a respondent in our studies, and teaching us what is important in old age
Teachers	Educational organization	Depending on the projects		Co-creating education and using our scientific insights in teaching their students

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
