# Peer review of "The Academic Collaborative Center Older Adults: A Description of Co-Creation between Science, Care Practice and Education with the Aim to Contribute to Person-Centered Care for Older Adults"

_ijerph, 2020, doi:10.3390/ijerph17239014_

Round 1

Reviewer 1 Report

The paper presents the assumptions regarding the strengthening of cooperation between science, education and long-term care of the elderly developed at Tilburg University.

The authors may consider following comments:

Major comments:

  1. The presented article is not a typical medical work. Instead of data comparisons and statistical calculations, the Authors decided to present a description of cooperation between a medical university and nursing homes, as well as the science that emerges on the basis of this co-work. The positive aspect of the work is the detailed description  of  the presented consortium structure,  funding,  process of emerging scientific topics et cetera  in the following chapters.
  2. The preparation of this type of work without statistical calculations requires the presentation of strong data supporting the correctness of the authors' actions. And while the described approach is absolutely correct from a social point of view, there is no scientific evidence of what the consortium has achieved in spreading its approach. I suppose as the end points "national inspiration day", "board game" and leaflets are given. In my opinion, it is not enough to in any way assess the scientific impact of the presented work.

Minor comments:

  1. Figure 1 is illegible. I cannot find a reference in the text to this graph.

2. Language, vocabulary requires  greater differentiation. I n some paragraphs there are a lot of repetitions. i.e.  paragraph 3.4  implementation is repeated 9 times. 

Summary:

As a geriatrician working at the university and cooperating with nursing homes, I must admit that I  started to read this paper  with great interest, and  finally the article  has disappointed me. In order for the work to be of full value,  Authors should include hard, tangible effects of the developed strategy. Otherwise, even the best-developed cooperation will not be evaluated. In  presented form, the work has no scientific credit and should be rejected.

Author Response

not applicable at this moment

Reviewer 3 Report

Since Edith Balint the concept of Patient Centered Care has developed in many areas, as opposed to the concept of Doctor centered care. Educational programs are traditionally integrated into care practice. The practical integration of scientific knowledge data beyond the closed world of academic researchers is an excellent initiative. 

The principles of this co-creation are well set out in the narrative methodology, but in the context of a peer review research paper, it lacks the means of evaluation to prove the "enhancing". An assessment tool and two practical outcome are well-cited primarily in doctoral studies, but we would like to know the practical result for the patient.

The design of the study is narrative with a description of an experience of person-centered care for older adults. The first word of the title "Enhancing" is unfortunately not demonstrated in the absence of an evaluation tool, results and discussion.

Round 2

Reviewer 1 Report

In the references section, the name of the first Author is mentioned 24 times. Why then, having so much expertise in the subject, the Authors are so modest in presenting their achievements? Only by reading works cited between numbers 21 and 35 gives a better outlook on the entire project. paragraph 2.3 is very poor in information what has actually been done.  On the contrary, I like the part  3.1. especially  276-287.  Authors have finally  given some information  about the results they have implemented so far.

What significantly changes the tone of the presented article is the emphasis on the future perspective. This approach is good. However keeping the article in convention “ look what we’ve done so  far,  and that’s only the preludium”  would have kept me reading the article with full admiration.

Line 284-  there is “ adultsand” and should be “adults and”

 In figure 1 I don’t  understand the idea of doubled boxes  as left and right are stating the same.  And some boxes are common for  independent living seniors and primary informal caregivers.

 Line 341 Double dot.

Line 345  Dot is missing.

Reviewer 2 Report

I would like to thank the authors for their response. All my comments have been sufficiently addressed in the manuscript. It is an easy to read and highly warranted paper. 

Author Response

Thank you for your valuable feedback and compliment that it is now an easy to read and highly warranted paper.

Reviewer 3 Report

Thank you for considering the suggestions. The corrections made allow a better understanding of the work carried out and its evaluation. The involvement and coordination of all healthcare professionals centered around the patient on the basis of validated scientific data is perfectly adapted to this type of care. Too often the doctor gives isolated medical indications, while delegating the therapeutic education of the patient to the other health professionals.

Author Response

Thank you for your valuable feedback.